# High-$Q$ TeO$_2$–Si Hybrid Microring Resonators

**Khadijeh Miarabbas Kiani** *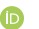**, Dawson B. Bonneville, Andrew P. Knights and Jonathan D. B. Bradley**

Department of Engineering Physics, McMaster University, 1280 Main Street West, Hamilton, ON L8S 4L7, Canada; bonnevd@mcmaster.ca (D.B.B.); aknight@mcmaster.ca (A.P.K.); jbradley@mcmaster.ca (J.D.B.B.)
* Correspondence: miarabbk@mcmaster.ca

**Abstract:** We present the design and experimental measurement of tellurium oxide-clad silicon microring resonators with internal $Q$ factors of up to $1.5 \times 10^6$, corresponding to a propagation loss of 0.42 dB/cm at wavelengths around 1550 nm. This compares to a propagation loss of 3.4 dB/cm for unclad waveguides and 0.97 dB/cm for waveguides clad with SiO$_2$. We compared our experimental results with the Payne–Lacey model describing propagation dominated by sidewall scattering. We conclude that the relative increase in the refractive index of TeO$_2$ reduces scattering sufficiently to account for the low propagation loss. These results, in combination with the promising optical properties of TeO$_2$, provide a further step towards realizing compact, monolithic, and low-loss passive, nonlinear, and rare-earth-doped active integrated photonic devices on a silicon photonic platform.

**Keywords:** silicon-on-insulator (SOI); silicon photonics; optical microrings; tellurium oxide

## 1. Introduction

The impact of silicon photonics on the development of photonic integrated circuits (PICs) is considerable, because of device compatibility with complementary metal–oxide–semiconductor (CMOS) technology and the leveraging of decades of research stimulated by the microelectronics industry [1]. The miniaturization of photonic waveguides has emerged as one of the most prominent technology platforms for PICs in recent decades [2].

The reduction in propagation loss associated with silicon waveguides is an ongoing research area because loss is a key parameter for link budget management in on-chip optical networks and many important micro-photonic devices, as well as passive and active systems including sensors, filters, delay lines, and light sources [3–6]. One device structure for which propagation loss is particularly critical is the microring resonator (MRR). MRRs provide an efficient cavity which has a compact size, wavelength selectivity, tunability, scalability, and functional versatility [7], making them a prominent candidate for a variety of applications including lasers [8,9], optical sensors [10], nonlinear optics [11,12], quantum optics [13], (de-)multiplexing systems [14], optical filters [15], and optical modulators [16]. The loss in MRRs can be quantified by the cavity $Q$ factor, which is typically limited to values on the order of $10^5$ in standard 220 nm-high single-mode silicon MRRs around 1550 nm.

Several approaches have been explored for enhancing the $Q$ factor in silicon MRRs, including minimizing light scattering at roughened sidewalls via shallow-etched or multi-mode waveguide designs and alternative fabrication methods based on selective oxidation, e-beam lithography, reactive ion plasma etching, inductively coupled plasma (ICP) etching, or low-pressure chemical vapor deposition (LPCVD). High-$Q$ factor silicon MRRs fabricated using a selective oxidation process have been shown with intrinsic $Q$ factors of $5.1 \times 10^5$ [17] and $7.6 \times 10^5$ [18], although challenges remain in the control of the fabrication process [19]. A silicon microring resonator with a $Q$ factor of $1.7 \times 10^6$ was demonstrated using a large cross-section multi-mode waveguide with a severely limited free spectral range (FSR) of 17.1 pm [20]. With a similar approach, a silicon microring resonator with a high $Q$ of $1.3 \times 10^6$ and larger FSR was proposed in [21], with a bend radius of 450 μm. A silicon MRR

with an internal $Q$ factor of $1.1 \times 10^6$ and FSR of 0.208 nm and utilizing a multi-mode ridge waveguide was fabricated using a standard CMOS-compatible silicon-on-insulator (SOI) process [22]. A multi-mode ultrahigh quality factor racetrack resonator with a 1.6 μm width was proposed using a standard single-etching process with a quality factor of $2.3 \times 10^6$ provided by a multi-project wafer foundry [23]. An internal quality factor of $2.2 \times 10^7$, corresponding to a 2.7 dB/m propagation loss, was achieved in a silicon microring resonator with a radius of 2.45 mm cladded with silicon oxide [24] by oxidizing the wafer surface in a steam oxidation process and using a reflowing photoresist strategy. An intrinsic $Q$ factor of $1.57 \times 10^6$, corresponding to a waveguide loss of 0.35 dB/cm, was realized in a silicon MRR with a radius of 150 μm and an FSR of 0.845 nm using e-beam lithography with a top cladding of a glass-like compound from hydrogen silsequioxane (HSQ) covered with a silicon oxide layer [25]. Low-loss submicron silicon-on-insulator strip waveguides were reported with a 0.5 dB/cm loss at 1310 nm and a 30 μm bend radius, cladded with silicon oxide [26] using $H_2$ plasma post-lithography treatment and $H_2$ thermal annealing after silicon etching. More generally, propagation losses lower than 0.4 dB/cm for the *C*-band and 0.8 dB/cm for the *O*-band of silicon wire waveguides have been reported for waveguides with a 440 nm core width, 220 nm core height, and 2 μm-thick $SiO_2$ cladding layer defined by a high-resolution immersion lithography process [27]. While all of these approaches to improving $Q$ factors in silicon MRRs are promising for different applications, they either suffer from performance trade-offs (e.g., multi-mode operation, larger footprint, and/or significantly reduced FSR) or add fabrication cost and complexity.

In this paper, we demonstrate a silicon MRR with a $Q$ factor of $1.5 \times 10^6$ at 1550 nm, corresponding to a propagation loss of 0.42 dB/cm, fabricated with a standard foundry process, plus a low-temperature post-process deposition of a $TeO_2$ cladding layer. In addition to enabling a straightforward and monolithic low-loss hybrid waveguide structure, $TeO_2$ has promising optical properties for new functionalities in silicon photonic microsystems. $TeO_2$ has been shown to be thermally and chemically stable, possess high nonlinearity, and low optical attenuation from visible to mid-infrared wavelengths (0.4~5 μm), and have a high refractive index (2.1 at 1550 nm) and low dispersion [28]. Furthermore, the unique site variability in the $TeO_2$ glass matrix enables high rare-earth dopant solubility and leads to large emission bandwidths, motivating its application in integrated optical amplifiers and lasers [29–31], including the recent demonstration of a hybrid rare-earth laser directly on silicon with an internal quality factor of $5.6 \times 10^5$ [32]. This low-loss platform has significant potential for linear, nonlinear, and active optical applications in silicon photonics.

## 2. Microring Resonator Fabrication and Design

The microring resonator structure used in this work is displayed in Figure 1. It consists of an integrated silicon microring and bus waveguide coated with a thin film of tellurium oxide ($TeO_2$). The silicon structure was fabricated in a silicon photonics foundry on a wafer-scale SOI platform with a 220 nm silicon layer thickness and consists of a 30 μm-radius silicon microring constructed using a 0.5 μm-wide waveguide. The bus waveguide is 0.4 μm wide, and the point coupling gap is 1.0 μm. The structure was cladded with $SiO_2$, and subsequently, a window on top of the microring resonator was etched to the silicon layer for use in the post-processing $TeO_2$ deposition or unclad device experiments. A set of devices with identical dimensions but without the $SiO_2$ cladding etched was also fabricated for comparison. Deep etching was used for end-facet preparation and wafer dicing. For the $TeO_2$-clad devices, the structure was coated with a 270 nm-thick $TeO_2$ film deposited using a room-temperature reactive RF co-sputtering post-processing step with 145 W of tellurium target sputtering power, 2.8 mTorr chamber pressure, and 12 and 7.6 sccm of argon and oxygen flow, respectively. The substrate temperature was set at 20 °C. A top-view scanning electron microscope (SEM) image and the cross-section diagram of the $TeO_2$-coated Si microring resonator are displayed in Figure 1a,b, respectively. Figure 1c,d show the image

of the experimental setup and a microscopic image of the microring resonator with an open window structure.

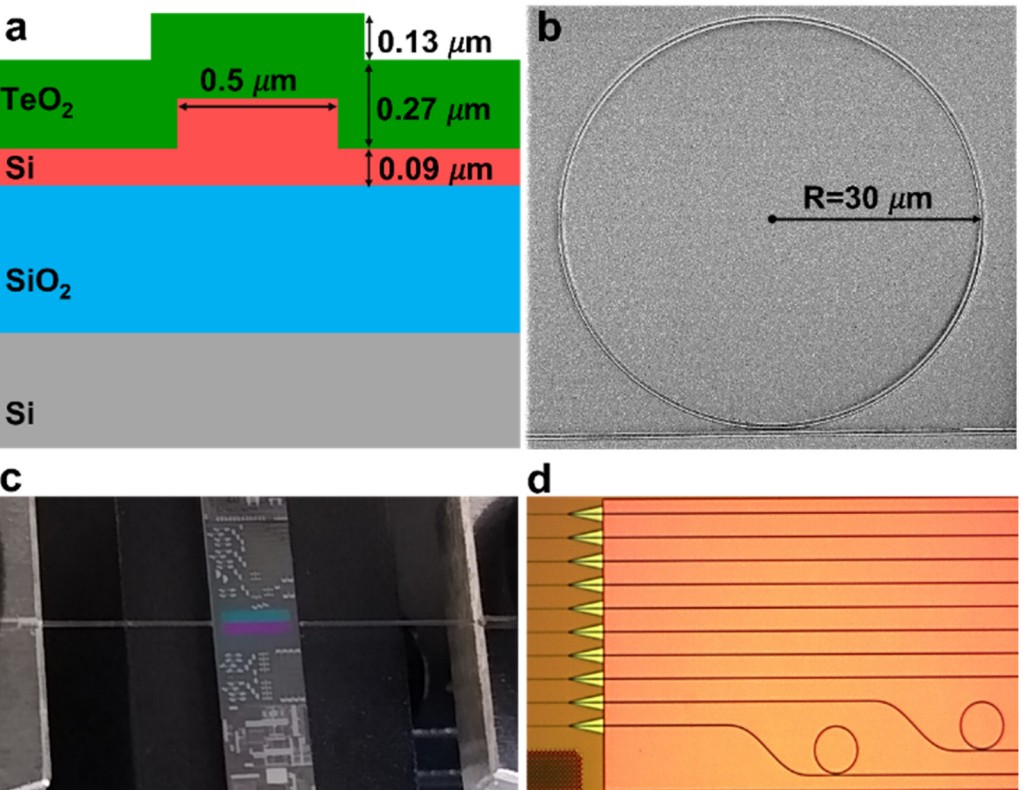

**Figure 1.** (**a**) Cross-section profile of the TeO$_2$-coated Si microring showing the microring structure. (**b**) Top-view SEM image of a TeO$_2$-coated Si microring resonator. (**c**) Photograph of the coupling setup during measurement showing the SOI chip with window opening for TeO$_2$ post-processing deposition. (**d**) Microscopic image of the open window microring resonator structure.

The electric field profiles of the transverse-electric (TE)-polarized fundamental modes calculated using a finite element method (FEM) modesolver for the (cladding–core) TeO$_2$–Si, SiO$_2$–Si, and air–Si waveguides at a 1550 nm wavelength are displayed in Figure 2a–c, respectively. We also summarize the calculated optical properties of the TeO$_2$–Si, SiO$_2$–Si, and air–Si microring resonator structures in Figure 2d. The microring resonators and the bus waveguides were designed to achieve single-mode waveguide conditions at 1.55 to 2 μm wavelengths. The ring waveguide structure supports the TE-polarized and TM-polarized modes, and it has a low bending radiation loss at 1550 nm for TE only. The effective index of the TeO$_2$-coated resonator is 2.8, which is almost 7.7 and 12% higher than that of the air- and SiO$_2$-clad devices. For the TeO$_2$-coated Si microring resonator, 21.7% of the optical power is confined in the TeO$_2$ coating layer, while 65.0% is confined in the silicon layer. The rest of the optical power is confined in the lower SiO$_2$ cladding. Slightly lower cladding confinement of 15.8 and 17.2%, respectively, is observed in the air- and SiO$_2$-clad cases. The effective area is slightly larger in the TeO$_2$-coated Si microring resonator than SiO$_2$-coated and uncoated silicon microring resonators, although the mode is pulled upward more into the cladding, both of which can influence the ring–bus waveguide coupling.

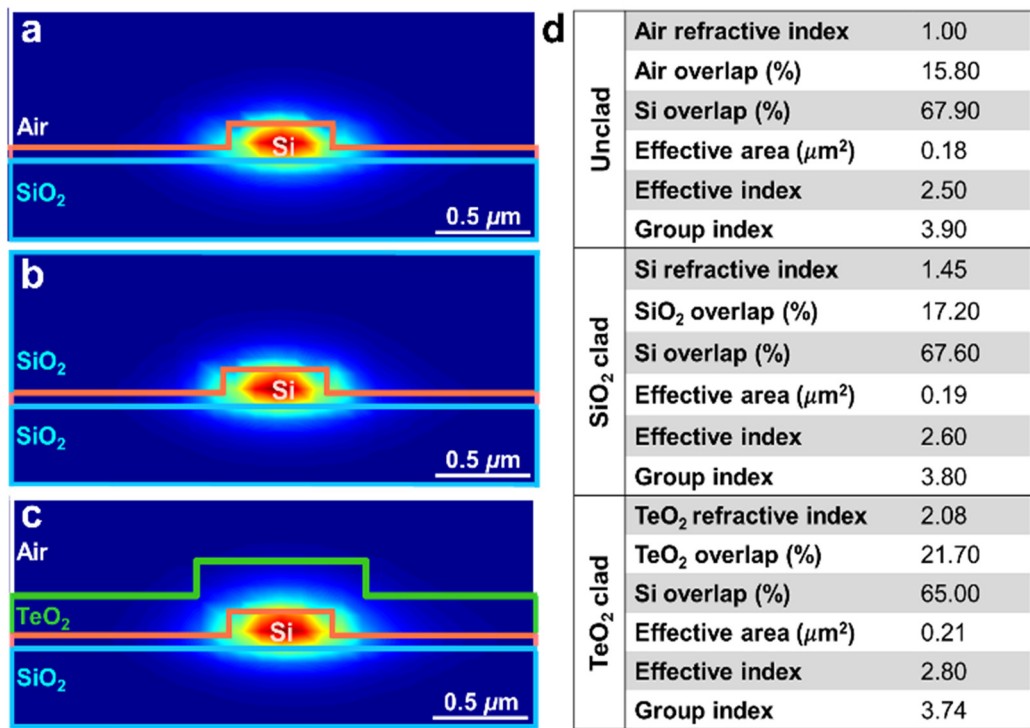

| | Air refractive index | 1.00 |
|---|---|---|
| **Unclad** | Air overlap (%) | 15.80 |
| | Si overlap (%) | 67.90 |
| | Effective area ($\mu m^2$) | 0.18 |
| | Effective index | 2.50 |
| | Group index | 3.90 |
| **SiO$_2$ clad** | Si refractive index | 1.45 |
| | SiO$_2$ overlap (%) | 17.20 |
| | Si overlap (%) | 67.60 |
| | Effective area ($\mu m^2$) | 0.19 |
| | Effective index | 2.60 |
| | Group index | 3.80 |
| **TeO$_2$ clad** | TeO$_2$ refractive index | 2.08 |
| | TeO$_2$ overlap (%) | 21.70 |
| | Si overlap (%) | 65.00 |
| | Effective area ($\mu m^2$) | 0.21 |
| | Effective index | 2.80 |
| | Group index | 3.74 |

**Figure 2.** Calculated electric field profile of the fundamental transverse-electric (TE)-polarized mode for (**a**) unclad, (**b**) SiO$_2$-clad, and (**c**) TeO$_2$-clad silicon microring resonators. (**d**) Calculated fractional optical intensity overlap factors and effective mode areas for the fundamental TE microring mode at 1550 nm wavelength.

The TeO$_2$-coated resonator has a higher mode area of approximately 0.20 $\mu m^2$ at 1.55 $\mu m$. The effective index increases for the TeO$_2$ film cladding as the resonant mode becomes more confined in the TeO$_2$ layer. The expansion of the optical mode at longer wavelengths decreases the effective index. The results show that approximately 60% of the mode power is confined in the Si region and 26% in the TeO$_2$-cladded layer for the TeO$_2$ cladding at a wavelength of 1.55 $\mu m$.

### 3. Microring Resonator Characterization

We used a fiber–chip edge coupling setup, a tunable Agilent 81640A 1510–1640 nm laser, and a fiber probe station to characterize the passive transmission properties of the silicon microring resonators. TE-polarized light from a 1550 nm tunable laser was launched into the chip through the polarization controller and lensed fiber, and the transmitted light across the chip was launched to the output lensed fiber connected to the Agilent power sensor. We observed TE single-mode resonances supported by the microring resonator, as shown in Figure 3a–c. For the TeO$_2$-coated silicon microring resonator, we measured a free spectral range (FSR) of 3.7 nm at a wavelength of 1573 nm. The air- and SiO$_2$-clad devices were under-coupled over the measured transmission range, while the TeO$_2$-clad MRR was under-coupled at shorter wavelengths and became critically coupled above ~1600 nm. By fitting the transmission responses of the under-coupled resonator using a Lorentzian function (as indicated in Figure 3d–f), we obtained internal quality factors, $Q_i$, of $2.0 \times 10^5$ at 1542.43 nm, $6.7 \times 10^5$ at 1587.28 nm, and $1.5 \times 10^6$ at 1579.94 nm for the uncoated, SiO$_2$-coated, and TeO$_2$-coated silicon MRRs, respectively, corresponding to 3.4 dB/cm, 0.97 dB/cm, and 0.42 dB/cm propagation losses in the microring [33]. The results are summarized in Table 1, including the TeO$_2$ film loss measured by prism coupling on a witness sample, showing low material loss, and fitted external and internal $Q$ factors for the MRRs with different top claddings.

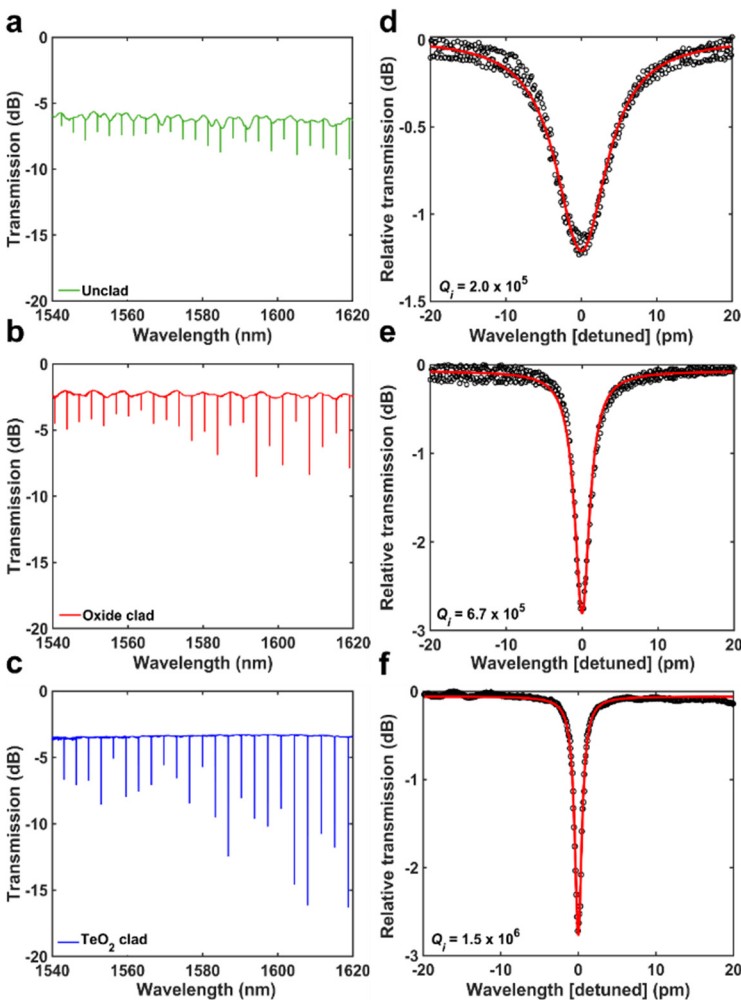

**Figure 3.** Measured TE transmission spectra for (**a**) uncoated, (**b**) $SiO_2$-coated, and (**c**) $TeO_2$-coated silicon microring resonators with a microring–waveguide gap of 1.0 μm. Close-up views of the under-coupled resonances for the (**d**) uncoated, (**e**) $SiO_2$-coated, and (**f**) $TeO_2$-coated silicon microring resonators showing an intrinsic quality factor, $Q_i$, of $1.5 \times 10^6$, corresponding to 0.42 dB/cm optical propagation loss for the $TeO_2$-coated silicon microring resonator.

**Table 1.** Measured properties of silicon microring resonators with different cladding materials.

| Cladding Material | Cladding Thickness (nm) | Film Loss @ 638 nm (dB/cm) | Film Loss @ 1550 nm (dB/cm) | Extinction Ratio (dB) | External Q Factor | Internal Q Factor | Propagation Loss (dB/cm) |
|---|---|---|---|---|---|---|---|
| Uncladded | —– | —– | —– | 1.2 | $2.8 \times 10^6$ | $2.0 \times 10^5$ | 3.4 |
| $SiO_2$ | 2000 | —– | —– | 2.8 | $4.2 \times 10^6$ | $6.7 \times 10^5$ | 0.97 |
| $TeO_2$ | 270 | $0.5 \pm 0.2$ | $0.1 \pm 0.1$ | 2.8 | $9.0 \times 10^6$ | $1.5 \times 10^6$ | 0.42 |

The total loss in the MRR can be expressed as the losses from the bulk materials, including contributions from the Si, $SiO_2$, and $TeO_2$ linear absorption and Si nonlinear (two-photon) absorption, radiation loss due to waveguide bends, and surface scattering losses related to the waveguide surface roughness and geometry. The bulk loss includes losses from impurities, internal defects, absorption loss due to chemical bonds, and nano- and micro-voids. Because of the high-quality silicon-on-insulator and $SiO_2$ foundry materials, and low $TeO_2$ film loss, we expected the bulk material loss contribution to be negligible. Furthermore, nonlinear optical loss was anticipated to be low due to the low power used in the transmission experiments.

We calculated the theoretical radiation loss and equivalent $Q$ factor for the $TeO_2$-coated silicon microring resonator structure using an FEM bent eigenmode solver, for varying bend radii, as shown in Figure 4a. The dashed lines in Figure 4a indicate the experimentally determined internal $Q$ factors for the MRRs with different claddings. The calculated radiation-limited $Q$ factors were measured to be $3 \times 10^8$, $1 \times 10^8$, and $1.7 \times 10^7$ for the $TeO_2$-clad, $SiO_2$-clad, and unclad microring resonators, respectively, at a radius of 30 μm, corresponding to 0.002 dB/cm, 0.007 dB/cm, and 0.04 dB/cm propagation losses. The results show that the radiation loss was low ($Q > 10^7$) at the selected bend radius of 30 μm.

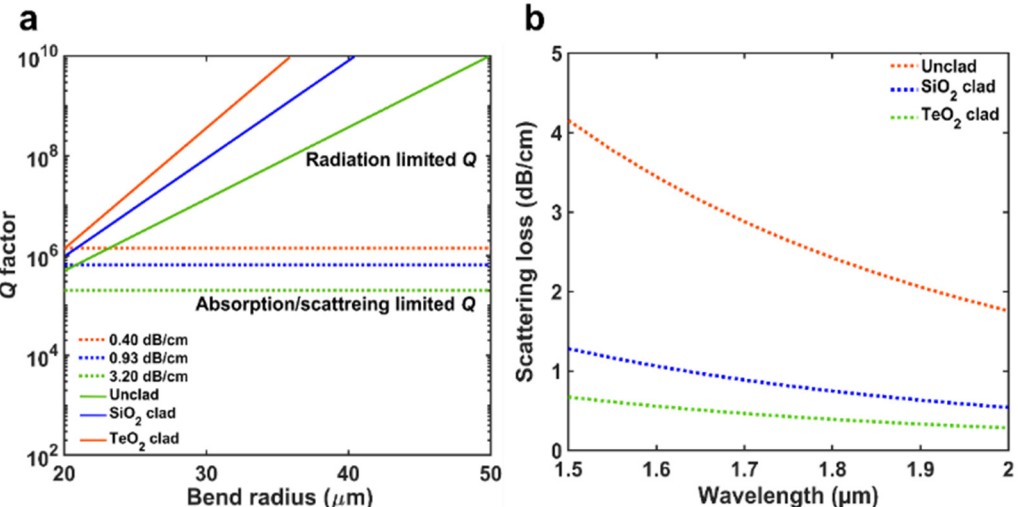

**Figure 4.** (**a**) Calculated internal Q factor of uncoated-, $SiO_2$-coated, and $TeO_2$-coated silicon resonator waveguides versus bend radius. (**b**) Calculated scattering loss as a function of wavelength for microring resonators with air, $SiO_2$, and $TeO_2$ claddings based on the 3D Payne–Lacey model.

The surface scattering loss is the waveguide scattering loss per unit length and was calculated using the widely used Payne–Lacey model [34]. This model has been applied previously to 3D-SOI bent waveguides [35–37] such that

$$\alpha_{Surface\ Scattering} = 4.34 \frac{\sigma^2}{\sqrt{2}k_0 \left(\frac{W}{2}\right)^4 n_1} f.g.\eta \tag{1}$$

where $\sigma$, $k_0$, $W$, and $n_1$ are the roughness, free-space wave vector, waveguide width, and core index, respectively. The function $f$ is determined by the correlation length ($L_c$), while g is determined by the geometry of the waveguide and accounts for ridge-type waveguide structures such as the SOI ridge rings investigated here. Both $\sigma$ and $L_c$ are typically measured via atomic force microscopy or scanning electron microscopy of a waveguide. We assumed values of 1.0 nm and 50 nm for σ and $L_c$, respectively, based on previous measurements on SOI waveguides [35–38]. We also assumed the roughness of the $TeO_2$ coating can be neglected based on previous AFM roughness measurements showing <1 nm root mean square (RMS) surface roughness on low-loss sputtered thin films [38]. Details of the expressions g and f can be found in the Payne–Lacey model [34]. The Payne–Lacey model was modified for bent waveguides and rings by adding a correction factor $\eta$ [35] which allows us to accurately predict the loss of bent waveguides or resonators given the value of the sidewall roughness. The correction factor $\eta$ is defined as the ratio of the mode overlap between bent and straight waveguides using an FEM simulation. In the Payne–Lacey bending model, $\eta$ is considered 1 in the straight waveguide. We plot the sidewall scattering loss as a function of the wavelength in Figure 4b. The experimental loss obtained from the quality factor measurements of the MRRs follows the same trend as the Payne–Lacey model, which indicates that the losses in the different MRRs are largely

influenced by the core cladding refractive index contrast and sidewall roughness [23]. These results show that the application of a high-refractive-index $TeO_2$ top cladding can be considered promising for low-loss passive components as well as active devices such as monolithic amplifiers and lasers with improved performance on the SOI platform [32].

### 4. Conclusions

In summary, we have demonstrated a compact, single-mode $TeO_2$–Si hybrid microring resonator with a $1.5 \times 10^6$ internal $Q$ factor corresponding to a 0.42 dB/cm propagation loss. The resonator was fabricated using a standard wafer-scale foundry process and a simple post-processing $TeO_2$ deposition step at room temperature. We propose that these results on tellurium oxide-coated silicon microring resonators are promising for the fabrication of integrated on-chip low-cost and high-performance rare-earth-doped active devices such as amplifiers and lasers, as well as low-loss passive and nonlinear devices in silicon photonic platforms, with signal processing, light generation, microwave photonics, environmental and biological sensing, and communications applications.

**Author Contributions:** K.M.K. designed the device; K.M.K. and D.B.B. laid out the silicon chips. K.M.K. developed the low-loss tellurium oxide films on the silicon chips. K.M.K. performed the experimental characterization and analysis. K.M.K. wrote the original manuscript. A.P.K. and J.D.B.B. reviewed and edited the text. A.P.K. and J.D.B.B. supervised the project. All authors have read and agreed to the published version of the manuscript.

**Funding:** This research was funded by the Natural Sciences and Engineering Research Council of Canada (grant numbers RGPIN-2017-06423, STPGP 494306, and RTI-2017-00474), the Canadian Foundation for Innovation (CFI project number 35548), and the Ontario Research Fund (ORF project numbers 35548 and RE-09-051).

**Institutional Review Board Statement:** Not applicable.

**Informed Consent Statement:** Not applicable.

**Data Availability Statement:** The data presented in this study are available on request from the corresponding author.

**Acknowledgments:** We thank CMC Microsystems and the SiEPIC Program for facilitating the silicon photonics fabrication, and the Centre for Emerging Device Technologies (CEDT) at McMaster University for support with the reactive sputtering system.

**Conflicts of Interest:** The authors declare no conflict of interest.

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
