# Peer review of "High-Q TeO2–Si Hybrid Microring Resonators"

_applsci, doi:10.3390/app12031363_

Round 1
Reviewer 1 Report
Dear Editor and authors:
I have read with interest and care the manuscript “High-Q TeO2-Si hybrid microring resonators”, submitted by Miarabbas Kiani and colleagues. The authors propose the use of tellurium-oxide-claddings to reduce the losses in SOI micro ring resonators (MRR) and thus attaining quality factors Q above 10^6 at 1.5 microns. After providing a comprehensive introduction to Si MRRs, the authors describe in detail the design and fabrication of the devices including the mode properties numerical assessment. Then, they perform a characterization of the transmission and losses characteristics, including the evaluation of the losses nature using the Payne-Lacey model for bent waveguides, concluding that the origin of the losses is mostly in the sidewalls roughness (scattering). The authors carry out as well the same characterization in air-coated and SiO2-coated Si MRR, and conclude that the overperformance of the TeO2 with respect to the formers comes from the increase in the material refractive index and its intrinsic material losses.
The authors have tackled a problem of great interest for the community working on Integrated Photonic Circuits and have chosen an ingenious solution when resorting to TeO2 claddings. This solution not only provides a larger refractive index but, as stated by the authors, it comes with an added functionality: the feasibility to generate light and thus fabricate integrated light sources. And this is a great plus. Their work is quite solid and thorough, as well as technically sound, concise, well-organized and very well written. Actually, I have no concerns regarding the content, and thus I am glad to recommend publication of this manuscript without further changes.
Author Response
Response to Reviewers
Manuscript ID: applsci-1573597
Manuscript Title: High-Q TeO2-Si hybrid microring resonators
"We thank the reviewer to read our work and we appreciate your very helpful and valuable comments and time."
Best Regards,
Khadijeh Miarabbas Kiani
Reviewer 2 Report
- “23.1% of the optical power is confined in the TeO2 coating layer” in line 117 and “TeO2 overlap 21.7%” in fig.2d, do they refer to the same quantity? Why are the values different although they all seem to represent intensity?
- What contributed to high propagation loss in uncladded case? (Table 1)
- Is the dashed lines in Fig.4a correlated with bend radius? And which result can reflect the conclusion “radiation loss is negligible (Q > 107) at the selected bend radius of 30 μm” mentioned in line 174.
- There is no comparison of the influence introduced by different sidewall roughness to come to the conclusion in line 195.
Author Response
Response to Reviewers
Manuscript ID: applsci-1573597
Manuscript Title: High-Q TeO2-Si hybrid microring resonators
We thank the reviewer for their very helpful and valuable comments and time. Please see the responses below.
- “23.1% of the optical power is confined in the TeO2 coating layer” in line 117 and “TeO2overlap 21.7%” in fig.2d, do they refer to the same quantity? Why are the values different although they all seem to represent intensity?
Thanks for this good comment. We noticed a typo error for this sentence as shown below (changes in red):
“For the TeO2-coated Si microring resonator, 21.7 % of the optical power is confined in the TeO2 coating layer, while 65 % is confined in the silicon layer.”
- What contributed to high propagation loss in uncladded case? (Table 1)
Thanks for this comment. Indeed, in uncladded cases, high refractive index contrast between the silicon and air causes this high propagation loss. One approach to low-loss optical waveguides, as we show here with the Payne-Lacey model, is reducing index contrast to minimize losses. This part has been mentioned in the text:
“The experimental loss obtained from the quality factor measurements of the MRRs follows the same trend as the Payne-Lacey model, which indicates that the losses in the different MRRs are largely influenced by the core cladding refractive index contrast and sidewall roughness [24].”
- Is the dashed lines in Fig.4a correlated with bend radius? And which result can reflect the conclusion “radiation loss is negligible (Q > 107) at the selected bend radius of 30 μm” mentioned in line 174.
Thanks for this great question. When bending loss becomes negligible (significantly less than other sources of loss) the internal Q factor of the resonator becomes limited by scattering and absorption losses. The scattering and absorption losses are relatively independent of bend radius for the bend sizes explored in this work, and are referred to here as the waveguide-limited internal Q factors for different waveguide propagation losses. We calculated the theoretical bend radiation loss and equivalent Q factor for coated and uncoated silicon microring structures using a finite element bent eigenmode solver, as shown in Figure 4a. The dashed lines in figure 4a indicate the internal Q factors corresponding to different absorption/scattering-limited microring propagation losses (“waveguide propagation losses”).
For the calculated bend radiation loss Q, we converted the simulated bend loss to an equivalent internal Q factor for the resonator versus ring radius, assuming bending radiation is the only source of waveguide loss, as shown in Fig. 4a. The calculated radiation limited Q factors were measured 3x108, 1x108, and 1.7x107 for TeO2 clad, SiO2 clad, and unclad microring resonators, respectively, at a radius of 30 µm corresponding to 0.002 dB/cm, 0.007 dB/cm, and 0.04 dB/cm pf propagation losses. As a result, the radiation losses are low at the selected bend radius of 30 μm in comparison with the experimental propagation loss.
Therefore, we added the following sentence to line 174 (changes in red):
“The calculated radiation limited Q factors were measured 3x108, 1x108, and 1.7x107 for TeO2 clad, SiO2 clad, and unclad microring resonators, respectively, at a radius of 30 µm corresponding to 0.002 dB/cm, 0.007 dB/cm, and 0.04 dB/cm of propagation losses. The results show that the radiation loss is low (Q > 107) at the selected bend radius of 30 µm.”
- There is no comparison of the influence introduced by different sidewall roughness to come to the conclusion in line 195.
Thanks for this good comment. We changed the following sentence and added reference (changes in red):
“The experimental loss obtained from the quality factor measurements of the MRRs follows the same trend as the Payne-Lacey model, which indicates that the losses in the different MRRs are largely influenced by the core cladding refractive index contrast and sidewall roughness [24].”
Sincerely yours,
Khadijeh Miarabbas Kiani

Reviewer 3 Report
- In the characterization part, details of the devices are missing, for example which was the tunable laser used?, model, company, range of tunability, power, and so on so forth.
- Which spectrometer was used for measuring the spectra?.
Author Response
Response to Reviewers
Manuscript ID: applsci-1573597
Manuscript Title: High-Q TeO2-Si hybrid microring resonators
We thank the reviewer for their time and valuable comments. Please see the responses below.
- In the characterization part, details of the devices are missing, for example which was the tunable laser used?, model, company, range of tunability, power, and so on so forth. Which spectrometer was used for measuring the spectra?
The laser sources used for measurements was an Agilent 81640A tuneable 1510-1640 nm laser source. Similarly, the output light was coupled into an Agilent 81640A power sensor connected to the computer by GPIB cable to record the transmission.
Therefore, we added the following sentence in the conclusion (changes in red):
“We used a fiber-chip edge coupling setup, tunable Agilent 81640A 1510-1640 nm laser and a fiber probe station to characterize the passive transmission properties of the silicon microring resonators. TE polarized light from a 1550-nm tunable laser was launched into the chip through the polarization controller and lensed fiber, and the transmitted light across the chip was launched to the output lensed fiber connected to the Agilent power sensor.”
Sincerely yours,
Khadijeh Miarabbas Kiani
